# Edible Films on Meat and Meat Products

**Dong-Heon Song, Van Ba Hoa, Hyoun Wook Kim, Sun Moon Khang, Soo-Hyun Cho, Jun-Sang Ham** and **Kuk-Hwan Seol** *

Animal Products Research and Development Division, National Institute of Animal Science,
Rural Development Administration, Wanju 55365, Korea; sdh8507@korea.kr (D.-H.S.); b1983@korea.kr (V.B.H.);
woogi78@korea.kr (H.W.K.); smkang77@korea.kr (S.M.K.); shc0915@korea.kr (S.-H.C.); hamjs@korea.kr (J.-S.H.)
* Correspondence: seolkh@korea.kr; Tel.: +82-63-238-7375; Fax: +82-63-238-7397

**Abstract:** In 2018, the worldwide consumption of meat was 346.14 million tonnes, and this is expected to increase in the future. As meat consumption increases, the use of packaging materials is expected to increase along with it. Petrochemical packaging materials which are widely used in the meat processing industry, take a long time to regenerate and biodegrade, thus they adversely affect the environment. Therefore, the necessity for the development of eco-friendly packaging materials for meat processing, which are easily degradable and recyclable, came to the fore. The objective of this review is to describe the application of natural compound-derived edible films with their antioxidant and antibacterial activities in meat and meat products. For several decades, polysaccharides (cellulose, starch, pectin, gum, alginate, carrageenan and chitosan), proteins (milk, collagen and isolated soy protein) and lipids (essential oil, waxes, emulsifiers, plasticizers and resins) were studied as basic materials for edible films to reduce plastic packaging. There are still high consumer demands for eco-friendly alternatives to petrochemical-based plastic packaging, and edible films can be used in a variety of ways in meat processing. More efforts to enhance the physiological and functional properties of edible films are needed for commercial application to meat and meat products.

**Keywords:** meat; packaging; substitutional plastic; biodegradable material; biopolymers

## 1. Introduction

Food packaging is a multidisciplinary area that encompasses food science and engineering, microbiology, as well as chemistry, and ignited tremendous interest in maintaining the freshness and quality of foods and their raw materials from oxidation and microbial spoilage [1]. In the food industry, the packaging is an important step for the production, storage and transportation of food [1]. Packaging is in direct contact with the food and intended to contain it from its manufacture to its delivery to the consumer, in order to protect it from external agents, from alterations and contaminations, as well as from adulteration [2]. The complete goals of the food packaging are to (i) suppress microorganism growth, (ii) keep their stability against environmental hazards and resist against oxidation, (iii) mask the unpleasant odors while preserving flavor, (iv) sustained delivery of nutrients, (v) help the filtration and accumulation of elements, and (vi) act as a sensor carrier [1–3].

Food packaging comprises an important portion of the food industry, and innovation in this field has been motivated mainly by consumer needs and preferences, as well as the need for long-distance transportation while maintaining the freshness of the packaged food without silage symptoms [1,4]. The packaging sector has become an important part of the global industry and constitutes 2% of the Gross National Products (GNP) for developed countries [1,5].

However, the food materials commonly used as packaging are one of the major solid wastes in the main cities in the world [2]. In this regard, in the last decade, there has been growing attention given to environmental pollution issues [1,2,6]. It is estimated that the production of these materials is around 300 million tons, in which the applications of

these petroleum-based synthetic plastics as packaging represent about 39.6% of the total demand [2,7,8]. However, these plastics have harmful effects on the environment which take a long time to completely decompose [8].

Global efforts are being made to avoid the use of plastic in food packaging. Since 2018, the UK has banned the use of plastic in the packaging of fresh food. Additionally, the EU has passed a ban on single-use plastics from 2021. Therefore, healthy and safe food free from synthetic chemicals has become one of the key challenges for food manufacturers and food businesses [8]. In order to reduce the environmental impact, bio-based packaging materials are an attractive alternative to reduce the use of non-degradable and non-renewable materials [2,8]. Consumers' increasing concerns for health and the environment have intensified the focus of researchers on the use of bio-based packaging materials as an alternative to synthetic plastic polymer(s) in food packaging [8]. Some biopolymers degrade in just a few weeks, whereas degradation of synthetic polymers takes several years, depending on the type and origin of the polymer [8]. Such biodegradable alternatives include polysaccharides such as agar, chitosan, cellulose, starch, pectin and the proteins' potential replacements to plastic polymers [2,6,8].

The biopolymers can meet consumer demands of natural and healthy foods, utilization of food industrial waste and decreasing the burden of plastic waste disposal [9]. Biopolymers have applications such as edible films and coatings, in a wide range of products [2]. The concept of edible films evolved from environmental concerns, the increasing burden of plastic waste disposal, utilization of food industrial waste and consumer demands of natural, nutritional and healthy foods [9]. Edible films are structures for wrapping or interleaving products, prepared in order to obtain a thin thickness (layer of material), from biological macromolecules, which act as a barrier to external elements (moisture, gases and oils), protecting the products and increasing their shelf life [8]. They have amalgamated the concept of food, preservation and packaging into a film that is biodegradable, edible, prevents moisture loss, color fading, lipid oxidation, off-odors, enhances shelf-life, and imparts functionality on meat, fish and derived products [9].

Meat is an important component of the modern diet around the world due to its protein, minerals, vitamins, and micronutrients content [9]. Meat is rich in essential amino acids and is a good source of essential fatty acids (such as linoleic, linolenic and oleic acids) [9,10]. Moreover, lean meat is a good source of iron that is essential for the synthesis of hemoglobin, myoglobin, and certain enzymes and includes riboflavin, niacin, thiamine, and other B-complex group vitamins [9]. In 2018, people worldwide consumed 346.14 million tons of meat and meat production is expected to continue to increase that in 2030; this number will increase by 44% to 453 million tons [11]. This means that an increase in production and consumption of meat and meat product inevitably leads to an increase in the use of packaging materials. In the future, in order to suppress environmental pollution, it is necessary to use eco-friendly packaging materials instead of plastic packaging materials.

Meat and meat products are perishable food commodities that require proper processing and handling for extending the shelf life along with refrigeration [9]. Chemical degradation in meat and meat products is primarily lipid and protein oxidation under the influence of composition, composition, air, light and processing temperature [12]. The growth and proliferation of microorganisms in meat and meat products cause physical, chemical and sensorial changes [9,10]. Thus, incorporating functional agents that can prevent/arrest/avoid/minimize deteriorative processes into packaging has come up as a promising approach towards green preservation of meat and meat products [9,10].

Various studies were carried out with active edible films in meat and meat products. The use of active edible films is an option for meat, providing the probability that a single product offers both the packaging and the necessary antimicrobial and antioxidant protection with natural and biodegradable components [2]. Therefore, this review describes the application of edible films with natural compounds with antioxidants and antibacterials from eco-friendly material packaging for meat and meat products.

## 2. Edible Film Production

The edible film production processes can be divided into dry and wet [11].

The dry process of edible film production uses heat pressing, extrusion, and molten casting methods without liquid solvents such as water or alcohol [13]. For the dry process, heat is applied to the edible and biodegradable materials to increase the temperature to above the melting point of the materials, to cause them to mold [13]. Therefore, the thermoplastic properties of the raw materials should be identified in order to design edible film manufacturing processes [13].

The wet process of edible film production uses solvents for the dissolution of raw materials, followed by drying to remove the solvent and form an edible film structure [13]. In this process, the selection of solvent is the most important factor in the total process. Since the edible film should be for human consumption, only water, ethanol, and their mixtures are used as solvents [13,14]. In the first step, all the ingredients of edible film-forming materials are dissolved or homogeneously dispersed in the solvents to produce raw material solutions, the second step is applying this to flat surfaces using a dipping roller, spreader, or sprayer, and the third step is drying this to remove the solvent, forming a film structure [13].

Figure 1 shows a simple process of making edible films from polysaccharide gums. The polysaccharide (e.g., gum) is first dissolved in water and a uniform layer of slurry is drawn on a smooth surface such as a glass plate. After it is then dried to the proper moisture or humidity level, and peeled (Figure 1) [15].

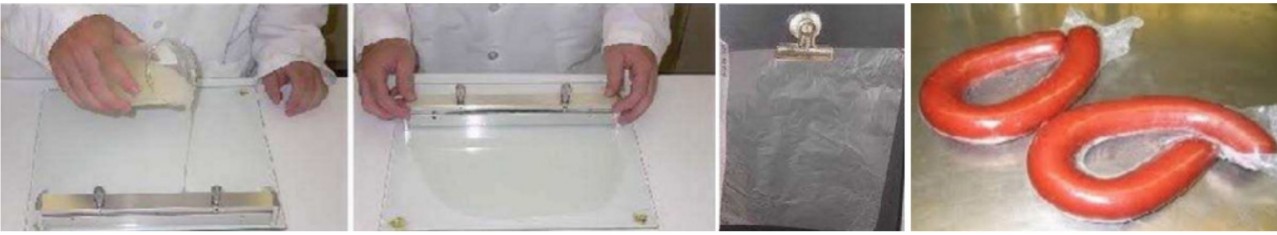

**Figure 1.** A simple lab process for making edible films from gum and examples of the sausage packaging. Reprinted from [15,16]. Copyright 2009 Springer.

## 3. Polysaccharide-Based Edible Films

Polysaccharides are widely available in nature and are non-toxic materials with selective permeability to carbon dioxide and oxygen [17,18]. Polysaccharides such as chitosan, pectin, gums, starch, cellulose, alginate, and carrageenan can confer good barrier properties to gases, in particular against oxygen, because of their ordered hydrogen bonds network, although most of these saccharides are sensitive to moisture due to their hydrophilic structure [19]. These properties of polysaccharides made them one of the materials that were used as a sustainable material in the formulation of coatings and edible films [17].

They are used in films to prevent dehydration, oxidative rancidity, and surface browning in food [19,20]. The polysaccharide-based films have high amylose starch appears to provide flexible and stretchable films [19,21]. However, due to their hydrophilic appearance in nature, they do not act as a barrier to moisture [22]. Because coatings consisting mainly of polysaccharides showed poor water vapor barrier characteristics, these coatings are considered to be a major factor affecting the decision of the marketers to not focus on delaying the moisture loss [22,23]. Other characteristics of polysaccharide coatings are that they are oil-free and colorless, which makes them have a small number of calories and can be applied to improve the shelf life of meat products, shellfish, vegetables, and fruits by considerably decreasing dehydration, oxidative rancidity, and surface darkening [22,23]. Table 1 shows an overview of the meat and meat product application of polysaccharide-based edible films and coatings.

**Table 1.** An overview of the meat and meat product application of polysaccharide-based edible films and coatings.

| Raw Material | Meat Product | Function | References |
|---|---|---|---|
| Cellulose | Minced camel meat | Antioxidation<br>Antimicrobial<br>Increased shelf life | [24] |
| Starch | Fresh beef | Antimicrobial<br>Increased shelf life | [25] |
| Pectin | Ham and sausage | Antioxidation | [26,27] |
| Gum | Fresh meat | Antimicrobial | [28] |
| Alginate | Beef steaks<br>Buffalo meat patties | Antioxidation<br>Antimicrobial<br>Inhibited discoloration<br>Decreased water losses | [29,30] |
| Carrageenan | Chicken breast | Antimicrobial<br>Increased shelf life | [31] |
| Chitosan | Dry fermented sausages | Antioxidation<br>Antimicrobial<br>Enhanced sensory attributes | [32] |
| | Cooked sausage | Antioxidation<br>Antimicrobial<br>Inhibited discoloration | [33] |

*3.1. Cellulose*

Lignocellulosic wood fibers comprise approximately 40–50% of cellulose and 25–30% of hemicelluloses by weight [34]. Cellulose is the most affluent organic compound on the earth [17]. Mostly, the cellulose is chemically altered during the dissolution process to facilitate the breakage of the polymer chains, and the cellulose derivatives are formed from d-glucose units linked by β-1,4 glycoside bonds [34]. Figure 2 shows the chemical structure of cellulose. The derivatives of cellulose are widely used in materials for edible films because they are biodegradable, tasteless and odorless [17,18,34]. Schantzet et al. [35] reported that the most used cellulose derivatives for making edible films are methylcellulose (MC), carboxymethyl cellulose (CMC), and hydroxypropyl methylcellulose (HPMC). These are eco-friendly materials that can form edible films with proper barrier and mechanical properties [17]. Cellulose derivative films generally possess high surface gloss, excellent transparency, good toughness and tensile strength [36,37]. In particular, the CMC was reported to exhibit excellent film-forming abilities which a water-soluble polymer and thermal gelatinization [38]. On the other hand, bioactive compounds may also be added to these films to introduce antioxidant and antimicrobial properties [17]. For instance, Nemazifard et al. [39] added pomegranate seed extract (PSE) to the CM, CMC, HPMC films to study the enhancement of antioxidant and rheological properties. The incorporation of PSE enhanced the antioxidant property of all cellulose derivatives films, although it reduced their barrier properties against oxygen [17,39]. Khezrian and Shahbazi [24] reported that the packed meats with nanocomposite-carboxymethyl cellulose films affect the retard protein (carbonyl content) and lipid oxidation (peroxide value and thiobarbituric acid reactive substances).

Hemicelluloses are amorphous and multifaceted heterogeneous polysaccharides that are structurally less ordered and are wood-based hydrophilic polysaccharides with a minimum thermal resistance [36,40,41]. Hemicellulose-based films are brittle; however, plasticizer addition could improve their flexibility, toughness as well as low oxygen permeability [42,43]. These films have an interesting role in packaging applications due to their low oxygen permeability. Both the softwood-hemicelluloses (galactoglucomanan and mannose) and the hardwood-hemicelluloses (glucuronoxylan) were reported in blends/composites with bioplasticizer to synthesize packaging materials with higher flexibility and oxygen permeability [44–46]. Though the films were vulnerable to aqueous uptake the higher added polymer content (alginate or CMC) increased mechanical strength by reducing film

flexibility and improved moisture uptake resistance [36]. A positive effect of methylcellulose coatings on meat is that they reduce oil absorption during fried cooking [47]. In the future, cellulose derivatives films need improvement because of their poor water vapor barriers due to their hydrophilicity nature [48].

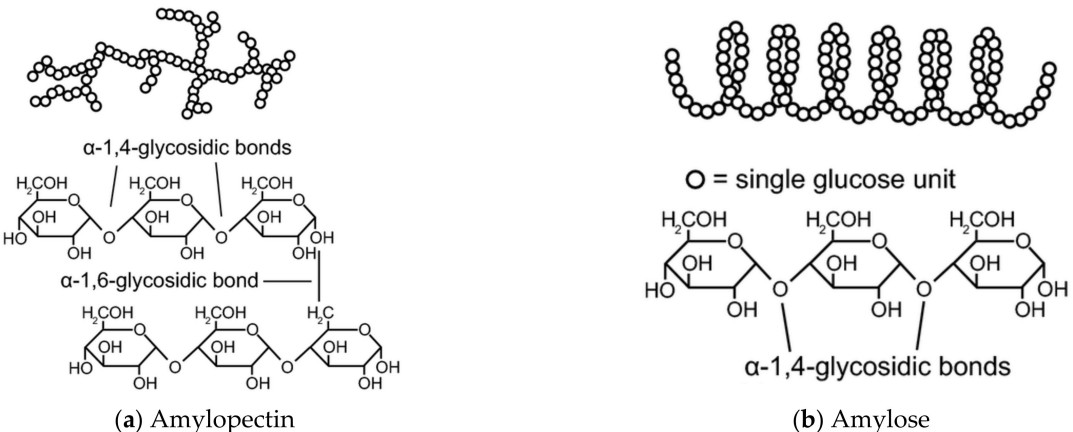

**Figure 2.** Chemical structure of cellulose. Reprinted from [49]. Copyright 2013 INASP.

### 3.2. Starch

Starch is the most common plant-derived polysaccharide that has been used for the development of bioplastic films due to cost-effectiveness, abundance, and significant film-forming properties [34,50,51]. A natural biopolymer synthesized from renewable resources is starch [52]. Since starch is cheap, readily available and totally biodegradable, there has been growing interest in synthesizing starch-based products [52]. The majority of biodegradable plastics are not commonly used since they are quite expensive, and the range of the material selection suitable for various end-use products is limited [52]. For wider industrial applications of biodegradable polymers, it is important to minimize their cost by blending them with renewable feedstock [52]. Blending biodegradable plastics with thermoplastics such as starch emerges to be a promising approach to reduce the cost of biodegradable polymers [52]. Starch is a natural polysaccharide that is used to make translucent or transparent, colorless, flavorless, and tasteless fabric biodegradable films or starch films [17,53]. Starch can be found in semi-crystalline hydrophilic granules form [17]. Figure 3 shows the chemical structure of starch, in which amylopectin is a branched polymer, on the other hand, amylose is a linear polymer forming a helix [54].

**(a)** Amylopectin                                                 **(b)** Amylose

**Figure 3.** Chemical structure of amylopectin (**a**) and amylose (**b**) from the corn starch. Reprinted from [54]. Copyright 2019 Elsevier.

The starch granules absorb water molecules surrounding the free hydroxyl groups, which makes the starch granules swell, then the swelling continues till a critical concentration is achieved [17]. The critical concentration is defined as the required concentration

of starch to make the swollen granules at 95 °C; on cooling a gel is formed [17,55]. The starch films generally exhibit a poor barrier property, however, incorporation of different nano-particles was reported to reduce the permeability of films to water vapor and gases [56]. The incorporation of nano-clay into starch could reduce the permeability of gases because the nano-clays provide the layers that are resistant to water vapor and oxygen permeation [56,57]. Different starches from previously unexplored resources, alone or in blended form with other natural polymers as biodegradable coatings or packaging films were reported to extend the product's shelf life [58–60]. A number of antimicrobial agents and starch are compatible, resulting in starch films that have the ability to deactivate a wide range of pathogenic bacteria [52]. Out of antimicrobial peptides, nisin or pediocin was incorporated into the starch-halloysite nanocomposite films via a casting method for food preservation [52,61]. These prepared films not only possess improved water barrier properties, thermal stability, and mechanical strength but also showed high antimicrobial activity against *C. perfringens* and *L. monocytogenes* [52,62]. Several studies have shown that incorporation of plant extracts and essential oils (e.g., rosemary extract, thymus kotschyanus, cassava, green tea, palm oil, oregano and black cumin) in starch films improves antioxidation and enhances oxygen barrier properties [56,63–66]. In meat packing, starches with plant-based (e.g., incorporation of cassava starch-containing kaffir lime leaves) edible coatings on beef had the effect of increasing antimicrobial content and better maintained beef quality based on pH and color than non-coated beef (Utami et al., 2017). Starch with essential oil (*Syzygium aromaticum*) biopolymeric packaging film, an antioxidant effect, use as active packaging for sausages [25]. Mohan et al. [67] showed that shelf life extension of mutton was possible through starch with spice-fused (*S. aromaticum* and *C. cassia*) edible film packaging. Therefore, the enrichment of starch with plant or starch with essential oil-based packings could extend the shelf life of meat and meat products, and also, could be used as an alternative to plastic packing methods [25].

Moreover, a decrease in water vapor permeability of composite film as compared with waxy corn starch and modified blends was also observed [58]. The modification of this exceptional biopolymer has found applications in the packaging and bioplastic sectors, as its applications have expanded to be utilized in biomedical, pharmaceutical, tissue engineering scaffolds, drug delivery, and even in metallurgical sectors for the development of porous media [60,68].

*3.3. Pectin*

Pectin is an anionic polysaccharide with a structural backbone of (1→4)-linked α-d-galacturonic acid unit (Figure 4) [17,20]. Pectin is a soluble element found in plant fiber and the plant's cell walls [22]. These plants—derived polysaccharides—are poor barriers to moisture and therefore, show better performance in foods containing low moisture [22]. It is used as a stabilizing, thickening and gelling agent in products such as; sausage, jams, milk, ice-cream, and yogurts [69,70]. Pectin edible films and their derivatives may be used in food packaging applications [17]. Giancone et al. [71] reported that pectin-based edible films exhibit excellent mechanical properties, an excellent barrier to oil and aroma, oxygen, high initial modulus, but they also show poor resistance to moisture, low elongations and are quite brittle; the addition of plasticizer makes them more flexible. Pectin films crosslinked with polyvalent cations (e.g., calcium) exhibit fair mechanical properties [72]. Films/gels of pectin are effective in the preservation of foods containing low moisture [73]. They are now used in the packaging of fresh meat, vegetables and fruits [74,75]. It is often seen that pectin is used as a source to improve the quality of meat patties (cooking loss, texture and antioxidation) [69]. Kang et al. [27] reported that lipid oxidation decreased and radical scavenging increased and the numbers of total aerobic bacteria were significantly reduced in cooked pork patty coated with pectin-based material in comparison with non-coating pork patty.

Figure 4. Chemical structure of pectin. Reprinted from [20]. Copyright 2018 Elsevier.

### 3.4. Gum

The use of plant gums in the food industry has a history of many decades [76]. Produced by different plants, all of them are polysaccharides with unique functionalities [76]. Gums obtained from plants either after the natural exudation process or employing extraction of tissues from different botanical parts are called vegetable gums [76,77]. Chemically, these are polysaccharides in which different monosaccharide units are joined through glycosidic linkages (Figure 5) [76,78]. Many of the exudate gums, such as acacia and karaya and seed/mucilaginous gums such as guar and locust bean, are known for their commercial applications all over the world [76]. Acacia species-derived Arabic gum is the most industrially employed polysaccharide because of its film-forming, encapsulation properties and unique emulsification [17,79]. The Arabic gum is composed of galactose, rhamnose, arabinose, and glucuronic acid [80], it has been used in food, cosmetics and pharmaceutical industries [81]. Additionally, meat coated with Arabic gum with garlic and cinnamon was effective in extending shelf life up to 3 weeks at 5 °C due to antibacterial activity [28].

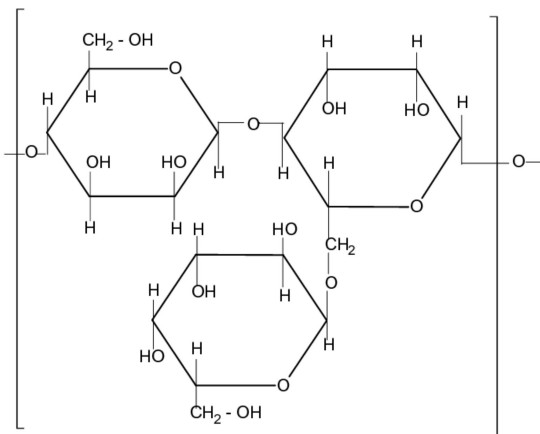

Figure 5. Chemical structure of guar gum. Reprinted from [82]. Copyright 2011 AJOL.

### 3.5. Alginate

Alginates, as a natural polysaccharide extracted from the brown algae of marine (Phaeophyceae), are compounded by L-guluronate (G) and R-D-mannuronate (M) units at various ratios and sequences in the (1→4) chain depending on the age of the plant and the source of alginate (Figure 6) [17,18,83]. By the addition of calcium ions, the formation of gels involving the G blocks is carried out, so the higher gel strength, the higher concentration of G units [17,84]. Removing the negative charge by cross-linking the alginate with calcium increased the tensile strength of the edible film, which the film structure forms better without the charge repulsion [15]. Alginate-based films can reduce meat shrink and improve juiciness and texture, but free calcium and metal cations needed to fix alginate coatings may induce undesired proteolytic activity [85,86]. Alginate-based coatings can preserve foods by increasing the water barrier, maintaining flavor, and retarding fat

oxidation [87]. Furthermore, alginate is kno wn to have enhancing effects on storage properties of meat and meat products by mixing with various biodegradable (e.g., ascorbic acid, carboxymethyl cellulose, chitosan, starch, calcium and essential oils) [15,29,88–91]. The alginate-based edible coatings decreased lipid oxidation of the meat compared to the non-coating meat, also, the alginate with oregano oil coating treatment showed the highest inhibited color losses, water losses, and lipid oxidation compared to the non-coating meat [29]. Keshri and Sanyal [30] researched to improve the quality of buffalo meat patties via dipping method during the end of the broiling process in alginate solution with preservatives for 30 s. In this method, improvements in tenderness, antioxidant and antimicrobial effects were shown on buffalo meat patties [30].

**Figure 6.** Chemical structure of sodium alginates. Reprinted from [83]. Copyright 2016 Taylor & Francis.

### 3.6. Carrageenan and Furcellaran

Seaweed is known as a prominent source of polysaccharides with properly established extraction processes, allowing to employ them in various areas [90]. Carrageenan is another polysaccharide that is sulfated water-soluble polymers extracted from various red seaweeds of the Rhodophyceae family [17]. Carrageenan consists of alternating β-(1, 4) and α-(1, 3) glycosidic bonds (Figure 7a–c) [15,17]. These polysaccharides are used in the dairy, pharmaceutical, and food industries as emulsifying, gelling, and stabilizing ingredients [91]. Karbowiak et al. [92] report that the mechanism underlying the carrageenan film formation involves the gelation during moderate temperature drying, leading to formed solid film after solvent evaporation by polysaccharide-double helices. The carrageenan edible films have widely been used for different purposes such as; sausage casings making, and for preventing superficial dehydration of dry solid foods, oily foods, meat, fish and poultry [92]. Seol et al. [31] reported that as effective in the antimicrobial activity of κ-carrageenan-based edible film containing ovotransferrin on fresh chicken breast was proved against *Escherichia coli*, *Staphylococcus aureus*, *Salmonella typhimurium*, and *Candida albicans*.

Furcellaran, as a new biopolymer, has been used in the development of films and coatings for food packaging applications [90,93]. Furcellaran is a naturally sulfated anionic polysaccharide typically obtained from the extract of red algae (*Furcellaria lumbricalis*), which possesses properties reminiscent of both agar and carrageenan [90]. Furcellarans are traditionally found in nature as a mixture of sodium, potassium, magnesium and calcium salts of a linear polymer, composed of (→4)-3,6-anhydro-d-galactopyranose-(1→3)-galactopyranose-4′-sulfate -(1→) structural units (Figure 7d) [90,94,95]. This is primarily because it does not present toxicity, is biodegradable, biocompatible, soluble in water and possesses an exceptional capacity to form gel [93,96]. Consequently, European Union (EU) regulations recognize the furcellaran is classified together with kappa, iota and lambda carrageenan as E407 because of those structural and functional similarities [93], and can be used as a food additive in the EU indicating that it is safe for human consumption [96].

**Figure 7.** Chemical structure of carrageenan and furcellaran. Reprinted from [97,98]. Copyright 2015 Ingenta and 2009 Woodhead. (**a**) Kappa carrageenan. (**b**) iota carrageenan. (**c**) Lambda carrageenan. (**d**) Furcellaran.

Furcellaran films present a slightly yellowish transparent appearance [90,93]. In many applications, transparent films are preferred as they can enhance the packaged product and influence consumer purchase intention [90]. The incorporation of green tea extract, yerba mate and ZnONPs made the yellow color more intense [90]. Furcellaran films with nanofillers (carbon quantum dots, maghemite nanoparticles and graphene oxide) indicated an inhibitory effect against the growth of *Salmonella enterica*, mainly because the polymer matrix contains reactive sulfate groups [90]. The nanocomposites of CQDs showed an inhibitory effect against *Staphylococcus aureus* (Gram-positive) and *Escherichia coli* (Gram-negative) growth [90]. Furcellaran can be one of the basic components in the production of "smart" films such as packaging materials with active and/or intelligent properties [90]. Packaging systems with bioactive compounds may have antimicrobial and/or antioxidant properties [90]. The active compounds may be released on the surface or absorbed into the packaged food products, leading to improved storage stability of the foods [90]. On the other hand, intelligent packaging is a material that monitors the quality of packaged food. This type of material is enriched with special indicators that provide qualitative information through visual colorimetric changes [99]. However, the use of furcellaranin on meat and meat product packaging is less common.

*3.7. Chitin and Chitosan*

The chitin is naturally found in the exoskeleton of crustaceans, cell walls of fungi, and other sources [17]. Chitosan is a deacetylated form of chitin, derived from the shell of shrimps and the cell wall of fungi, and is made up of N-acetyl-D-glucosamine residues joined by β-(1–4) glycosidic (Figure 8) [19]. This substance exhibits high solubility under acidic conditions and presents antimicrobial activity when the pH value is lower than its pKa (6.2–7.0) [19,100,101]. This polycationic polymer was tested as a carrier for active substances such as organic acids, essential oils, and derivative organic compounds [19,100].

**Figure 8.** Chemical structure of chitosan. Reprinted from [102].

Chitosan is a high molecular weight cationic polysaccharide that exhibits a great film-forming capacity, and antimicrobial activities [17,103,104]. Chitosan film was used as a packaging material for the preservation of different foods particularly when it is combined with other film-creating materials [17,105]. Coating food with chitosan films lowers the oxygen partial pressure in the package, keeps the temperature with moisture transfer between food and its environment, controls respiration and declines dehydration [17]. Moreover, chitosan is used for deacidification, setting texture, enhancing the emulsifying effect, natural flavor, and color stabilization of foods [106]. Chitosan-based films are clear, flexible and tough [17], well resistant to fat, oil, oxygen, but highly sensitive to moisture [17,107]. The antimicrobial activity of chitosan depends on several factors such as its molecular weight (MW), degree of deacetylation (DDA), conformational structure, interactions between positive charges (C-2 position of the glucosamine monomer) of the polymer, and anionic components of the bacteria cell surface such as sialic acid in phospholipids which limit the flow of microbial substances resulting in the membrane leakage [19,32,108]. High MW of chitosan forms a shield and prevents the movement of nutrients through the cell wall in Gram-positive bacteria, whereas its ability to attach to Gram-negative bacteria is enhanced when its MW is low, causing flocculation and disruption of physiological processes [101]. High DDA values, which are obtained from alkaline treatment around 80 °C, enhance the antimicrobial activity of chitosan due to stronger electrostatic interaction between the positive charge of this compound and the negative charge of the cell surface [100].

A recent study on Alaskan pollock sausages reported that films made up with chitosan–gelatin matrix inhibited both Gram-positive and Gram-negative bacteria growth in the product during 42 days of storage, and this antimicrobial effect is attributed to the hydrophobic amino acid residues of the chitosan [109] reported interesting results on suppression of fungal growth during ripening when chitosan was applied to fermented sausages, achieving around a two log cycle decrease in the mould and yeast count. Fungal proliferation on casings of sausages was also mitigated when chitosan at a 1% level was applied, exhibiting same effectiveness as potassium sorbate at 20% [32].

On the other hand, the antioxidant action of chitosan is associated with the chelation of free iron and other essential ions driven by thermal treatment of meat, which may destabilize the microbial cell wall and interfere with their enzymes [110]. Chitosan can also modify the oxygen barrier through interactions with ionic functional groups that restrict the PMF, though the antioxidant effect of this polysaccharide is strongly related to its molecular weight, and thereby, to its concentration [19,32,111].

The advantages of coatings made of chitosan with oregano for sausages was shown in pork sausages, in which lipid oxidation in the coated meat product (9.64 mg TBARS/kg) was lower than in the uncoated sample (29.23 mg TBARS/kg) at 15 °C and 75% relative humidity (RH) for 150 days. Moreover, no statistical changes in color were found between the tested products [111]. Additionally, Arslan and Soyer [32] found that TBARS values at the end of the ripening period of sausages were significantly lower for those treated with chitosan solutions in comparison with the control group. The mechanism behind the antioxidant effect of the chitosan-based films may be attributed to the chelating of free iron released during the storage of meats from myoglobin during the spoilage process [56,112]. Additionally, the chitosan's free amino groups can scavenge free radicals [113]. A study on litchi fruit demonstrated that chitosan coatings reduced polyphenol oxidase activity during storage [112].

Lekjing [33] evaluated peroxide values and TBARS in cooked pork sausages wrapped with films containing chitosan and clove oil compared to those in which the last one was not present in the coating, both stored at 4 °C. All samples showed an increase in the mentioned parameters, but the product covered with 2% chitosan and 1.5% clove oil had the lowest values [33]. Furthermore, the L* value of control samples increased, whereas the mixture composed of chitosan and clove oil exerted antioxidant activity over the sausage, which is related to the TBARS value [33]. This phenomenon was related to the synergistic

effect of the mixture, in which chitosan act as a chelator of ions that promote oxidative reactions in lipids [19].

## 4. Protein-Based Edible Films

Fibrous and globular proteins can be found naturally, in which the globulars are rolled over their selves, and the fibrous ones are bonded to each other in parallel [17,114]. The lactic serum, caseinate, collagen and seine are the proteins that can be used in edible films [17]. The mechanical properties of protein films are better compared to that of the polysaccharides films due to their unique structure [17]. When compared to synthetic polymers, the protein films generally have a good gas barrier and mechanical properties and poor water vapor permeability [115–117]. Figure 9 schematically shows the preparation of the protein film [117]. Figure 10 shows a visual appearance of the edible films on protein application [118]. Otoni et al. [117] reported that reduction of droplet size and addition of emulsions in isolated soy protein film generally decreased water vapor permeation in the films as well as total pore volume.

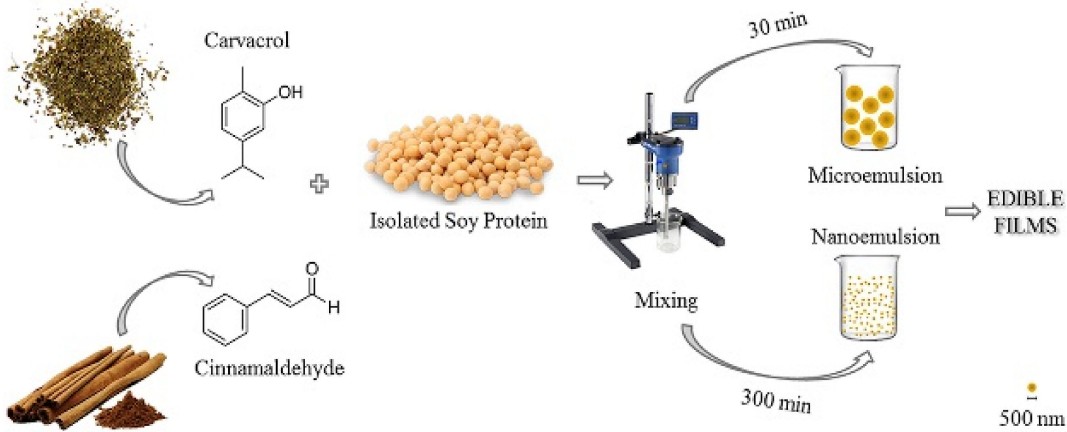

**Figure 9.** Manufacture of isolated soy protein containing carvacrol and cinnamaldehyde composite edible films. Reprinted from [117]. Copyright 2016 Elsevier.

(**a**) Whey protein film      (**b**) Whey protein with *L. casei* film

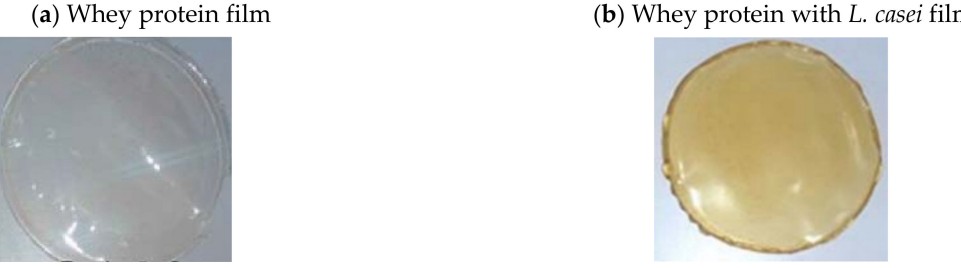

**Figure 10.** Visual appearance of the protein-based edible films at only whey protein (**a**) and whey protein with *L. casei* (**b**). Reprinted from [118]. Copyright 2019 AIDIC.

Functional properties of protein-based films depend on the degree of extension of protein chains, and the sequence of residual amino acids. Some materials such as collagen, keratin and proteins from quinoa, milk, egg, whey, zein, and soy are widely used [119]. Denatured proteins of the three last sources allow to obtain more cohesive structures than polysaccharides, and their adherence to hydrophilic meat surfaces helps to create a barrier against oxygen and carbon dioxide [19]. Nonetheless, protein-based films may exhibit low resistance to both mechanical stress and water diffusion, and they may become fragile under dried conditions [19]. In addition, the use of these macromolecules is a topic for discussion because they may be susceptible to native enzymes, or cause adverse

reactions on consumer metabolism due to allergenic protein fractions [120,121]. Table 2 showed an overview of the meat and meat product application of protein-based edible films and coatings.

**Table 2.** An overview of the meat and meat product application of protein-based edible films and coatings.

| Raw Material | Meat Product | Function | References |
|---|---|---|---|
| Casein | Salami | Antimicrobial | [122] |
| Whey protein | Chicken breast | Antimicrobial Antioxidation | [123] |
| Gelatin | Beef | Antimicrobial Inhibited discoloration Antioxidation | [124] |
| | Minced pork | Antimicrobial Increased shelf life | [125] |
| Soy protein | Fresh beef patties | Antimicrobial Antioxidation | [126] |
| | Chicken breast | Improved the flavor and texture | [127] |

### 4.1. Milk Protein

Milk protein contains casein and whey protein. Milk proteins can form flexible, flavorless and transparent films [17]. Casein forms edible films from aqueous caseinate solutions [17]. The obtained casein solution is then washed, dissolved in the alkali to increase the pH value to 7, and is finally dried. If $Ca(OH)_2$ is used, calcium caseinate is formed [17]. Since calcium cations help promotes cross-linking protein to protein interactions, calcium caseinate films have better barrier properties but it is more rigid [128]. On the other hand, if NaOH is used to increase pH value, a sodium caseinate is formed (Mohamed et al., 2020), and the sodium caseinate films usually have optical and tensile properties [128].

Whey protein is obtained after the precipitation of casein protein [17]. Whey proteins contain several materials, such as α-Lactalbumin, β-Lactoglobulin, immunoglobulins, bovine serum albumin and protease peptones [114]. They also serve as a carrier of food additives, antioxidants, colorants and antimicrobial agents [129,130]. The general form of whey protein, used to form edible films, is whey protein isolate (90% protein) [17]. For the fabrication of films, the first step is to get a concentrated solution of proteins over which heat is applied to denaturalize the proteins [17]. After this, it gets refrigerated to eliminate the enclosed gas and obtain the packaging material. These edible films are a better oxygen barrier at low or intermediate RH, nevertheless, they have poor water vapor permeability [17]. Whey protein has exceptional functional characteristics and film-forming abilities [34]. Films based on whey protein were examined for their excellent transparency, flexibility, oil/gas barrier properties at relatively low humidity [34,131]. However, whey protein-based films show poor water barrier characteristics [132]. The incorporation of essential oils/lipids was shown to reduce the drawback of the whey-based films [131,133]. However, determination of interaction between various biopolymers during coatings formation is necessary in order to develop packaging material with desired characteristics and functionality [134] Çakmak et al. [135] developed edible films based on whey protein isolate plasticized with glycerol and reported that bergamot essential oils and lemon acted as the active ingredients in the bioplastic films. These authors also found an excellent antimicrobial potential, significant oxygen permeability and water vapor permeability when incorporating essential oils. [135].

Whey protein concentrate (25–80% protein) is another type used in the past to form edible films, nevertheless, it may contain other impurities (e.g., lactose) that can enhance water vapor permeability, but worsen the mechanical properties [136]. Calcium caseinate and whey protein concentrate generally are weaker than whey protein isolate. Whey protein isolate can replace up to 50% of calcium caseinate without reducing the puncture

strength in making the edible films [137]. The lactic serum is a good barrier to $CO_2$ even though it is fragile. For solving this problem, a plasticizing agent (e.g., glycerol) could be used to enhance its mechanical properties [17].

### 4.2. Collagen and Gelatin

Collagen is obtained from mammals' skin or fish skin and bone. For decades, collagen edible films have been used in meat products to reserve humidity and give a uniform feature to the products [17]. In animals, the collagen and gelatin contents constitute about 20–25% of total body mass [34]. Its structure consists of three cross-linked α-chains while the denatured collagen derivative is called gelatin that is composed of many polypeptides and proteins [34]. Collagen is constituted mainly by methionine, hydroxyproline/proline, and glycine amino acids [138]. Collagen-based bioplastics are synthesized by the extrusion process and comprise various applications, while gelatin-based film production requires a wet process by the formation of a film-forming solution [34,139]. Collagen-based bioplastic films generally exhibit good mechanical properties for instance; hydrolyzed collagen films were reported to possess excellent tensile strength [140]. However, the gelatin films usually exhibit poorer mechanical and barrier properties [141].

Gelatin is produced by hydrolysis of collagen [63]. On heating, collagen film acts as an edible skin and it assists the meat product cooking [142]. Collagen, mainly present in animal skins, muscle, bones and connective tissues, is pre-treated with an acid/alkaline solution, and further heating up to 40 °C gives gelatin [17,143]. Pure and dry gelatin is transparent, tasteless, brittle, odorless and glass-like solid, with faint yellow color [143]. The gelatin-derived edible film is usually made by dissolving the gelatin solution in hot water and then casting it on a plate and finally drying it in an oven [144,145]. When the protein content is increased, the gelatin-based edible films have higher film thickness and better mechanical properties, but it may also result in decreased water vapor permeability [146]. The antioxidant properties of the pure gelatin and gelatin/chitosan-based films were examined by the scavenging 2, 2-diphenyl-1-picrylhydrazyl (DPPH) radicals and ferric to ferrous reducing power assays [147]. These authors reported that gelatin alone and gelatin/chitosan (75:25% ratio) films exhibited higher antioxidant properties than the other compositions studied.

### 4.3. Soy Protein

Soy protein is a plant protein derived from soybeans. Nowadays, soy protein is available in the markets in different forms such as; soy flour (56% protein and 34% carbohydrate), soy concentrate (65% protein and 18% carbohydrates), and isolated soy protein (90% protein and 2% carbohydrates) [148]. Isolated soy protein is usually used to produce soy protein film [17]. This is attributed to the fact that the non-protein fraction in other soy protein forms negatively affects film formability [149]. Soy protein edible films are generally produced by Yuba films or by baked film methods [17]. In the Yuba film method, soy milk is boiled in a thin pot, resulting in formed surface films which are then dried in air. In the baked film method; spread soy protein isolates are baked on baking pans for 1 h at 100 °C [150,151]. In general, soy protein films are smooth, flexible and clear in comparison to the films formed from other plant sources-derived proteins [152]. For the gas barrier properties, the soy protein films are superior to that of lipids and polysaccharides. Their oxygen permeability, if they are not exposed to moisture, is at least 260 times lesser than those made with low-density polyethylene, starch and pectin [153]. Soy proteins are also exploited for bioplastic films synthesis for packaging applications [34]. Films based on soy proteins are smoother, transparent, flexible and cost-effective than other protein-based bioplastics [117]. Moreover, they also exhibit good oxygen barrier properties under low moisture conditions [152]. However, major disadvantageous points of using these soy protein films are low mechanical strength and heat stability, and allergenicity as compared to the low-density polyethylene (LDPE) [34].

## 5. Lipid-Based Edible Films

Lipids are molecules that are commonly found in natural sources (e.g., animals and plants) [17]. The diversity of the lipid functional groups is made up of phospholipids, phosphatides, mono-, di- and tri-glycerides, terpenes, cerebrosides, fatty alcohol, and fatty acids [17,52]. Fats and oils are mixtures that are found in animals and plants, respectively, and their major constituents are triglycerides. In terms of chemical structure, both of them are similar to each other but they vary physically, as fat and oil are often seen in the solid and oil forms, respectively at room temperature. [17]. The manufacture of lipid-based films is called larding [85]. The food industry has been paying attention to the application of lipids in coatings and edible films production for the preservation of foods. Lipids in coatings and edible film exert many features for instance; they provide gloss, minimize moisture loss, reduce cost, and complexity of packaging [17,154]. Rodrigues et al. [155] made a film using palm fruit oil and reported a high water resistance, water vapor barrier, transparency and elongation properties of the film. Vargas et al. [156] reported that coating pork meat hamburgers with sunflower oil film enhanced the quality of the products because the coating prevented the products from undesirable reaction during storage.

Many waxes and oils obtained from petroleum were used to create moisture barriers, and improve the cohesiveness and flexibility of films [19]. The main compounds used to produce these films are usually natural waxes such as beeswax and paraffin wax; resins; acetoglycerides; fatty acids; and both mineral and vegetable oils [120]. In meat products, sensory properties, such as tenderness, were improved by using lipid coatings [120,121]. On one side, caution should be exercised in their use as these films may give an undesirable taste to meat and meat products [19]. Table 3 showed an overview of the meat and meat product application of lipid-based edible films and coatings.

**Table 3.** An overview of the meat and meat product application of lipid-based edible films and coatings.

| Raw Material | Meat Product | Function | References |
|---|---|---|---|
| Essential oils | Fresh sausage | Antimicrobial | Araújo et al. [157] |
| | Fermented Sausage | Antimicrobial Inhibited discoloration quality preservation | Catarino et al. [158] |
| Emulsifiers | Yao meat products | Antimicrobial Antioxidation Improved quality and shelf life | Liu et al. [159] |
| Plasticizers | Chilled meat | Moisture barrier | Kyshenia et al. [160] |

### 5.1. Essential Oils

Essential oils were found to exhibit excellent antioxidant and antimicrobial activities [161]. Essential oils are secondary metabolites produced by aromatic plants which may have antioxidant effects, medicinal properties and fragrance [19]. Some examples are the alkyl group in the case of limonene and p-cymene; some functional groups and the presence of delocalized electrons in most terpenoids; and free hydroxyl groups and substitutions on the aromatic ring in phenylpropenes [162]. They contribute to important antimicrobial effectiveness owing to their ingredients from terpenoids, terpenes, and other aromatic compounds [163]. Their antimicrobial activity depends on their chemical composition and the number of single compounds, and their efficacy relies on the growth stage of spoilage microorganisms [164]. However, due to its stronger odors, the utilization of essential oils as food preservatives may be restricted [165]. The most studied essential oils were extracted from eucalyptus [166], *Syzygium aromaticum* [167], rosemary and thyme [168]. Although the capacity of these substances to efficiently reduce the microbial population present in the meat matrix was demonstrated, a high concentration of essential oils may modify the sensory properties of sausages due to their intense aroma [169–171]. In addition, Benbettaïeb et al. [172] have stated that the rate of release of bioactive compounds should be similar, or slightly faster, than the growth rate of pathogenic or spoilage microorganisms.

Therefore, it is necessary to design controlled-release mechanisms of these composites from the film onto food, considering the microbial growth kinetics [173].

In order to overcome this problem, edible films should be made with bioactive agents to introduce desired functionality [174]. The addition of other biodegradable materials to enhance the essential oil film gives better function. Randazzo et al. [175] used citrus essential oils of peels from lemon, mandarin, and orange in methylcellulose or chitosan films. Additionally, Du et al. [176] used cinnamon, allspice, and clove bud essential oils. In general, the films made with these essential oils showed excellent antibacterial activity without defects in tensile and water vapor properties. Araújo et al. [157] have demonstrated that the antimicrobial activity of essential oils is a function of the amount of oxygen available in packed meat. In this way, low oxygen diffusion avoids oxidative reactions in oils, and anaerobic microorganisms become more sensitive to the toxicity of these substances.

On the other hand, Portuguese sausages packaged in active coatings made of whey protein and Origanum virens essential oil were stored for 4 months, and they were preserved for 20 days (paínhos) and 15 days (alheiras) more in comparison with uncoated samples, allowing them to have a permissible microbiological count [158]. In the same way, the shelf life of sausages packaged in coatings with 2% chitosan and 1.5% clove oil was 10 days longer compared to a control (without coating), and 2% chitosan coating. This behavior was attributed to the presence of eugenol from clove oil, which inhibits the production of cellular enzymes and hence, induces cell damage [158].

*5.2. Waxes*

Waxes originated from vegetal and animals have a function of protective covering tissues. Waxes have a higher molecular weight because they are formed by alcohol and/or esters of a long-chain acid [17]. These are used for food coatings or edible films making for reducing moisture permeability in food products [154,177–179]. Saucedo-Pompa et al. (2007) designed an edible coating with Aloe Vera gel and Candelilla wax, applied on fresh-cut fruits, and these authors reported that Candelilla coatings were an alternative for preservation of fruits because it reduced weight loss, and enhanced firmness, lightness and appearance values compared to non-coated fruits. Kowalczyk [180] created films with 5% (*w/w*) aqueous biopolymer solutions containing 0.5% (*w/w*) Candelilla wax, 3% (*w/w*) sorbitol, and 0.35% (*w/w*) Tween 40 for carrier ascorbic acid. Spotti et al. [181] mixed beeswax, Brea gum (from *Cercidium praecox*), and glycerol, but they decided that beeswax did not help in this film because of decreased water vapor permeability, microstructure, and mechanical properties. However, Chiumarelli and Hubinger [182] designed a film with cassava starch, carnauba wax, glycerol, and stearic acid that presented amazing properties (e.g., good mechanical, barrier, physical, thermal, and structure properties). Research on the use of wax in edible film for meat products will need to be supplemented.

*5.3. Emulsifiers*

Emulsifiers are macromolecular stabilizers with ionic character which can decrease the surface tension between two immiscible phases at their interface, allowing them to become miscible [17]. The major function of emulsifiers is preventing phase separation since they keep hydrophilic–lipophilic balance [17,114,183].

Out of the emulsifiers, lecithin originated from vegetal (e.g., soybean lecithin) or animal (e.g., (egg lecithin), in the mixture or fraction of phospholipids form, is applicably important [17,184]. The soy lecithin film or coatings has beneficial effects on color stability, solubility, opacity and microstructure of foods. Andreuccetti et al., [185] observed small globules on the surface of highly concentrated lecithin film, and the authors indicated heterogeneity in the network of protein.

*5.4. Plasticizers*

Plasticizers are characterized by low migration and low toxicity, and they are used in combination with biodegradable polymers to enhance the mechanical properties of edible packaging materials [19]. Plasticizers are low-molecular-weight materials that can increase the strength and flexibility of edible film [17]. Their low molecular weight allows them to accommodate easily within intermolecular spaces of polymeric chains. However, low energy, vapor pressure and diffusion rate are required for the plasticizer to solvate and remain in the polymeric structure [19]. Additionally, after a critical concentration of plasticizer, phase separation may be observed [186]. The incorporation of plasticizers into coating or film can help to increase permeability to gases and water due to their capacity of intermolecular forces reduction in a polymer [17]. Polysorbates and glycerol are known as popular plasticizers [183,187]. The addition of diverse plasticizers (e.g., glycerol-sage seed gum) into coating or film was reported to show a positive effect such as; increased moisture content and thickness [188]. Jouki et al. [189] reported that morphology, glycerol-cress seed gum edible films were smooth and homogeneous without cracks.

The main role of plasticizers is to improve flexibility, reduce brittleness, and decrease both porosity and tendency to crack. A high concentration of hydrophilic plasticizers may increase water uptake of the polymeric structure, meanwhile, hydrophobic plasticizers can reduce its moisture permeability by closing micro-voids [21,186]. Sorbitol, mannitol, and polyethylene glycol are plasticizers frequently used in edible films, but polyols such as glycerol are the most common due to their stability and compatibility with hydrophilic compounds [21,119,158].

*5.5. Resins*

Resins are substances that are produced from plant cells to respond to injury or infection in trees and shrubs. Otherwise, they can be produced by some insect species for instance; *Laccifer lacca* produces shellac resin [17]. The majority of resins are translucent with yellowish-brown tones and physically solid or semisolid [190]. Chauhan et al. [191] and Chitravathi et al., [192] used shellac in edible coating and they applied it on green chilies and tomatoes. Those films exhibited quick drying nature, transparency, glossiness, and sound emulsion stability, especially the coating could prevent senescence and extend shelf life by 12 days [192].

## 6. Application of Edible Film Packaging for Meat and Meat Products

Meat is distinguished by tissue structure and is frequently treated by different processes that may result in an increased risk of contamination with microorganisms [17]. Due to huge and diverse manufactured meat and poultry products, different methods to manage food-borne pathogens and to prolong their shelf life are needed [193]. In recent years, antimicrobial and intelligent packaging has emerged as a food-safety hurdle technology [17].

Edible films and coatings were also reported to have good mechanical properties, gas and moisture barriers, and when applied on meat and meat products, they generally show a positive effect on sensory perceptions of the products [120]. In the meat industry, the application that has been used is by foaming, dipping, spraying, casting, brushing, individual wrapping, or rolling [194]. Recently, various forms of combined meat preservers were fabricated using bioactive edible films and coatings [17].

## 7. Conclusions

In the meat processing industry, the current consumer demand for high-quality, low-cost, sustainable, natural, and safe food packaging instead of non-biodegradable and non-renewable petroleum-based polymers and plastics is a scientific challenge. The applications of edible films in meat packaging technology provide an eco-friendly alternative to petrochemical-based plastics. This can solve the waste accumulation problem due to petrochemical-based plastics which are non-biodegradable in nature. Edible film for meat

processing can be used in a variety of ways, including polysaccharides, chitosan, animal and plant proteins, and lipids. Efforts are focused on developing the right combination of blended materials because the effectiveness of edible films for meat products depends on biopolymer types and bioactive compounds to enhance their functional properties. Further studies are needed to improve the properties of edible film to obtain properties similar to petroleum-based plastics.

**Author Contributions:** Conceptualization, D.-H.S. and K.-H.S.; formal analysis, D.-H.S. and K.-H.S.; validation, V.B.H., S.M.K., S.-H.C. and J.-S.H.; investigation, D.-H.S., S.M.K., S.-H.C. and J.-S.H.; resources, D.-H.S., S.M.K., S.-H.C. and J.-S.H.; data curation, D.-H.S. and V.B.H.; writing-original draft preparation, D.-H.S. and K.-H.S.; writing-review and editing, D.-H.S., V.B.H., H.W.K. and K.-H.S.; visualization, D.-H.S.; supervision, K.-H.S.; project administration, D.-H.S. and K.-H.S.; funding acquisition, K.-H.S. All authors have read and agreed to the published version of the manuscript.

**Funding:** This research work was supported by (2021) the RDA Fellowship program of the National Institute of Animal Science, Rural Development Administration and "Cooperative Research Program for Agriculture Science and Technology Development (Project No. PJ01492003)" Rural Development Administration, Korea.

**Institutional Review Board Statement:** Not applicable.

**Informed Consent Statement:** Written informed consent has been obtained from the patients to publish this paper.

**Conflicts of Interest:** The authors declare that there are no conflicts of interest in this research article.

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
