# Peer review of "Edible Films on Meat and Meat Products"

_coatings, doi:10.3390/coatings11111344_

Round 1
Reviewer 1 Report
I have following suggestions for authors.
- All Tables mentioned in manuscript are not in manuscript, please add all tables.
- I would recommend to improve the structure of review - first describe only the different types of films/packaging and than write chapters focused on different properties of meat and meat products packed in these types of films/packaging, which were analyzed. E.g. - chapter Textural properties; chapter microbial contamination; chapter oxidation processes etc. Because the title of the manuscript is not so well supported by the content. There are a lot of chapters about packaging (where we can find only a few information about application on meat) and only one chapter called Application of Edible Film Packaging for Meat and Meat products. So the previous suggestion should improve the manuscript.
Author Response
Reviewer 1 comments:
I have following suggestions for authors.
Thanks for your comments. The authors appreciate the editor’s and reviewers’ comments and for the assistance in preparing more accurate manuscript. Specific steps taken to implement the suggestions of the reviewers are detailed below.
Detailed comments:
1) All Tables mentioned in manuscript are not in manuscript, please add all tables.
All tables have been revised to be included in the manuscript.
Table 1 was rewritten in p 4 (submitted versions L156~158).
Table 2 was rewritten in p 13 (submitted versions L493-495).
Table 3 was rewritten in p 16 (submitted versions L616~618).
2) I would recommend to improve the structure of review - first describe only the different types of films/packaging and than write chapters focused on different properties of meat and meat products packed in these types of films/packaging, which were analyzed. E.g. - chapter Textural properties; chapter microbial contamination; chapter oxidation processes etc. Because the title of the manuscript is not so well supported by the content. There are a lot of chapters about packaging (where we can find only a few information about application on meat) and only one chapter called Application of Edible Film Packaging for Meat and Meat products. So the previous suggestion should improve the manuscript.
The authors have additional written about the benefits of meat packaging for different types of edible film (e.g. beneficial effects of antioxidant, antimicrobial, decreased water losses, etc.).
2-1) Line 180-182 and Line 194-197 in the revised document.
- The effect of the cellulose film on meat packaging
2-2) Line 242-254 in the revised document.
- The effect of the stach film on meat packaging
2-3) Line 277-281 in the revised document.
- The effect of the pectin film on meat packaging
2-4) Line 299-301 in the revised document.
- The effect of the gum film on meat packaging
2-5) Line 319-329 in the revised document.
- The effect of the alginate film on meat packaging
2-6) Line 346-349 in the revised document.
- The effect of the carrageenan film on meat packaging
2-7) Line 346-349 in the revised document.
- The effect of the carrageenan film on meat packaging
All authors appreciated the reviewer’s comments and suggestions once again. We revised our manuscript based on the comments.
Reviewer 2 Report
The review entitled Edible Films on Meat and Meat Products to deal with an interesting and a nowadays trend topic. Generally, it is well written and organized. However, before being accepted for publications I suggest a major revision.
General comments
The title of the review is edible Films on Meat and Meat Products, but besides the description of the various biomolecules for the development of the film, there is a poor description of the application on meat and meat products. I suggest revising the title of the review or focus more on the meat products as reported in the title.
Section 2
1)Include a figure in which the chemical structure of the mentioned polysaccharides is resumed
2)Add a table (with relative references) in which the methodologies to process the polysaccharides to make edible films is resumed
3) I suggest improving the part related to the alginates and in particular focusing on the derivatives
Section 3
1) Add a table (with relative references) in which the methodologies to process the mentioned protein to make edible films is resumed
2) Add images of examples of protein-based films
Section 4
1) Add a table (with relative references) in which the methodologies to process the lipid to make edible films is resumed
Author Response
Reviewer 2 comments:
The review entitled Edible Films on Meat and Meat Products to deal with an interesting and a nowadays trend topic. Generally, it is well written and organized. However, before being accepted for publications I suggest a major revision.
Thanks for your comments. The authors appreciate the editor’s and reviewers’ comments and for the assistance in preparing more accurate manuscript. Specific steps taken to implement the suggestions of the reviewers are detailed below.
General comments
The title of the review is edible Films on Meat and Meat Products, but besides the description of the various biomolecules for the development of the film, there is a poor description of the application on meat and meat products. I suggest revising the title of the review or focus more on the meat products as reported in the title.
The authors have additional written about the benefits of meat packaging for different types of edible film (e.g. beneficial effects of antioxidant, antimicrobial, decreased water losses, etc.).
1) Line 180-182 and Line 194-197 in the revised document.
- The effect of the cellulose film on meat packaging
2) Line 242-254 in the revised document.
- The effect of the stach film on meat packaging
3) Line 277-281 in the revised document.
- The effect of the pectin film on meat packaging
4) Line 299-301 in the revised document.
- The effect of the gum film on meat packaging
5) Line 319-329 in the revised document.
- The effect of the alginate film on meat packaging
6) Line 346-349 in the revised document.
- The effect of the carrageenan film on meat packaging
7) Line 346-349 in the revised document.
- The effect of the carrageenan film on meat packaging
Detailed comments:
Section 2
1)Include a figure in which the chemical structure of the mentioned polysaccharides is resumed
The chemical structure of the polysaccharide has been revised to include a resumed figure.
(1) Line 108-132 in the revised document, Figure 1. A simple lab process for making edible films
(2) Line 199: The structure of cellulose is described.
- Figure 2. Chemical structure of cellulose.
(3) Line 217-220: The structure of starch is described and a figure of the chemical structure is included.
- Figure 3. Chemical structure of amylopectin and amylose from the corn starch.
(4) Line 282-284: Figure 4. Chemical structure of pectin.
(5) Line 302-304: Figure 5. Chemical structure of guar gum.
(6) Line 330-332: Figure 6. Chemical structure of sodium alginates.
(7) Line 364-367: Figure 7. Chemical structure of carrageenan and furcellaran.
(8) Line 367-398: Figure 8. Chemical structure of chitosan.
2)Add a table (with relative references) in which the methodologies to process the polysaccharides to make edible films is resumed
We have newly written in Session 2 that an overview of methodologies to edible film production.
- Line 108-132 in the revised document; Added edible file prodution to session 2
3) I suggest improving the part related to the alginates and in particular focusing on the derivatives
The description of the alginate film has been additionally write.
- Line 313-315 and Line 319-329 in the revised document.
Section 3
1) Add a table (with relative references) in which the methodologies to process the mentioned protein to make edible films is resumed
Added a session 2 on overview to making edible film.
Also, Added example images for the production of protein-based films.
- Line 467-478: Figure 9. Manufacture of isolated soy protein containing carvacrol and cinnamaldehyde composite edible films.
2) Add images of examples of protein-based films
Added example images of protein-based films.
- Figure 10. Visual appearance of the protein based edible films at only whey protein.
Section 4
1) Add a table (with relative references) in which the methodologies to process the lipid to make edible films is resumed
We have newly written in Session 2 that an overview of methodologies to edible film production.
- Line 108-132 in the revised document; Added edible file prodution to session 2
All authors appreciated the reviewer’s comments and suggestions once again. We revised our manuscript based on the comments.
Round 2
Reviewer 1 Report
Accept in present form. The authors improve the manuscript according to suggestions.
Reviewer 2 Report
I recommend for publication